# From Linear to Circular Ideas: An Educational Contest

**Denner Deda** [1], **Murillo Vetroni Barros** [2], **Constança Rigueiro** [3,*] and **Margarida Ribau Teixeira** [4,5,*]

1 ISISE—Institute for Sustainability and Innovation in Structural Engineering, Department of Civil Engineering, University of Coimbra, 3030-790 Coimbra, Portugal

2 Laboratório de Estudos em Sistemas Produtivos Sustentáveis, Universidade Tecnológica Federal do Paraná (UTFPR), Ponta Grossa 84017-220, Paraná, Brazil

3 Institute for Sustainability and Innovation in Structural Engineering, Instituto Politécnico de Castelo Branco, 6000-084 Castelo Branco, Portugal

4 CENSE—Center for Environmental and Sustainability Research, Universidade do Algarve, 8005-139 Faro, Portugal

5 CHANGE—Global Change and Sustainability Institute, Universidade do Algarve, 8005-139 Faro, Portugal

* Correspondence: constanca@ipcb.pt (C.R.); mribau@ualg.pt (M.R.T.)

**Abstract:** This work proposes a framework with which to analyse Higher Education Institution (HEI) students' knowledge and understanding of circular economy (CE) concepts and the potential of the CE to promote sustainability, using a contest. The framework integrates CE principles and business models with sustainable indicators, and it was applied to the accepted projects by the contest jury. The contest was launched in 2021 by the CE Working Group of the Portuguese Sustainable Campus Network to encourage creativity and the development of CE projects at HEIs. HEIs can play an essential role in promoting environmental education and creating partners with new visions for society and the economy concerning sustainability, developing knowledge, values, attitudes, and behaviours regarding the CE. The projects were mostly based on the recovery of secondary raw materials/by-products, the CE business model, and the CE principle of value optimisation. In addition, a strong relationship with environmental indicators was observed, but social and economic indicators of the CE were only marginally considered by the students. Therefore, students considered the CE as mostly mainly being related to product recovery/optimisation; thus, the CE concepts and principles and their relationship to sustainability implementation require reinforcement and transversal approaches to increase this knowledge and its dissemination.

**Keywords:** circular economy; higher education institutions; sustainable development; university cooperation; educational context

## 1. Introduction

The circular economy (CE) concept challenges the current linear model of production and consumption. The circular concept redefines growth, focusing on positive society-wide benefits, gradually decoupling economic activity from the consumption of finite resources, and designing waste out of the system [1]. In closed-loop biological and technical schemes, all systems are designed to be regenerative so that materials flow back into the cycle after use to boost resource productivity and system sustainability [2]. Thus, a CE calls for rethinking production processes and developing sustainable business innovations to support the systemic changes towards a circular and sustainable economy [3]. This concept has emerged as a worldwide political vision in recent years; for example, China [4], Europe [5], and Australia [6] have all launched sustainability initiatives over the last few years. The transition toward a more circular economy is also integrated into the sustainability agenda of the United Nations, contributing to several of the 17 Sustainable Development Goals (SDGs), such as SDG2 (zero hunger), SDG 6 (clean water and sanitation), SDG 7 (affordable and clean energy), SDG 8 (decent work and economic growth), SDG 12 (responsible consumption and production), SDG 15 (life on land), and SG17 (partnerships for the

goals) [7–9]. The responsibility to facilitate the transition to a CE is shared by businesses, policymakers, and citizens, where active cooperation between actors is imperative [10]. For this reason, innovation is required, and it must be applied to individuals, teams, and organisations. This is crucial for the development of knowledge, values, attitudes, and behaviours [11].

In this context, Higher Education Institutions (HEIs) play an important role, since they contribute to economic progress and social wellbeing through knowledge creation and dissemination (research and teaching) and community development (outreach activities) [12]. Salas et al. [13] investigated the role of HEIs in the transition to a CE in Latin American countries and concluded that HEIs are vital stakeholders and critical actors for implementing CE and innovation systems and that they are relevant to any regional development. In addition, HEIs provide appropriate environmental education, with a high impact on training and preparing the future generation for a green society [14]. HEIs need to become more aware of environmental and social challenges, working independently as future experts to create sustainable and resilient business opportunities, and a sustainable public sector, encouraging society to move towards sustainable development [15]. Therefore, HEIs are in a position to facilitate CE learning by (i) teaching, researching, and reaching out to improve sustainability and circularity in project development, and incorporating these principles across disciplines; (ii) educating and training the next generation of professionals, which will have a decisive impact on their professional contexts and social engagements; (iii) promoting an institutional culture of sustainability, which will increase the awareness of university staff, as well as that of the local and broader communities; and (iv) implementing sustainable campus practices (e.g., reducing greenhouse emissions, establishing zero-waste production, promoting biodiversity, using energy and water efficiently, and reducing the ecological footprint) [3,12]. Furthermore, HEIs can contribute to the CE transition through partnerships with local businesses and industries, with mutual benefits, and by introducing CE frameworks directly into local and regional agendas [16].

The role of HEIs is categorised into five analytical categories, as proposed by the EMF [17] and referenced by Serrano-Bedia and Perez-Perez [18]: (1) teaching for CE; (2) leading innovation by students; (3) stimulating research on the CE; (4) leading and influencing local change; and (5) campus management. These authors concluded that most of the CE initiatives developed by HEIs have focused on curricula reforms, which is in accordance with the few studies found which analysed the practical implementation of a CE in the HEI sector [18]. Whalen et al. [19] used the game *In the Loop* to provide experiential learning for CE education. The study concluded that the game can be a helpful tool for CE education since students recognised the significance of the various actors in the production process and the importance of adopting business strategies, as CE concepts, to address material criticality concerns [19]. Mendoza et al. [3,20] proposed a methodological framework and guidelines to help HEIs apply CE thinking to sustainable campus management. They used the University of Manchester as a case study. They found CE barriers similar to those related to sustainability management in HEIs, namely weak commitment and resistance to change, lack of strategic leadership and support from senior managers, conservative organisational structure and governance, limited specialisation, training, and capability of staff, poor communication, lack of data collection systems and appropriate performance indicators, and few incentives and financial resources [20]. Bugallo-Rodríguez and Vega-Marcote [21] studied the impact of a set of activities designed and implemented to improve the attitudes and actions of students to reduce their daily environmental impact on campus (Faculty of Educational Studies of the University of Coruña, Spain) and to be active agents for change regarding the CE. The methodology included three activities where the students explored, analysed, and solved particular problems related to the issues addressed. The results demonstrate that the activities caused the students to reflect and act on their daily impacts, but they did not apply the CE principles [21]. These studies present applied research activities implemented in HEIs to disseminate and promote CE. They emphasise the underdeveloped HEI research activities

and industry collaborations to increase the impact of teaching and research, and campus initiatives to implement the CE mentality and eco-responsible citizenship, as concluded by Serrano-Bedia and Perez-Perez [18]. In this context, the Portuguese Sustainable Campus Network proposed a contest for HEI students that challenged them to create and develop innovative projects to implement a CE on an HEI campus. The aim of the contest was to disseminate the CE concepts through innovative projects across Portugal and Portuguese Language Countries (CPLP) HEI campuses. This study aimed to evaluate the impact and students' knowledge of the CE concepts and their potential for application on HEI campuses, based on the contest. The contest was launched in 2021 by the working group of the CE of the Portuguese Sustainable Campus Network.

The present study aimed to answer the following research questions:

■ Do the students understand and know the concepts behind the CE?
■ Can the students develop innovative projects, including CE concepts?

This study contributes to the development of the CE by promoting leading innovative projects and helping HEIs in the implementation of a CE mentality in HEI students. It supports new visions beyond the HEI lecture halls, contributing to increasing HEIs' role and challenges in developing a CE.

## 2. Methodology

The present research was based on a four-step process aimed to evaluate the impact and students' knowledge of the CE concepts and their potential to be applied on a Portuguese-speaking HEI campus (Figure 1). The contest evaluation was backgrounded in CE models and indicators and was developed by the authors to achieve the stated goal of the research. This contest is presented as a case study where the students' perceptions and knowledge of CEs were analysed.

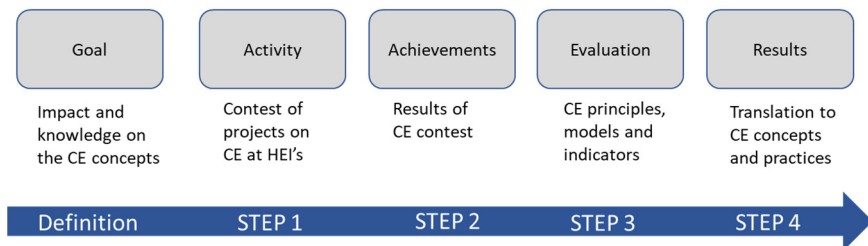

**Figure 1.** The methodology applied in the study.

### 2.1. The Contest—A Tool to Promote the CE in the HEIs (Steps 1 and 2)

#### 2.1.1. Contest Framework

The first circular economy competition, entitled "From linear to circular ideas-FL2CI", started in March 2021. This competition was created by the CE Working Group of the Sustainable Campus Network, Portugal, to promote and develop ideas and solutions to CE challenges in HEIs that could be applied on higher education and Portuguese-speaking institutions' campuses. The competition was intended to encourage the students to think and reflect on what a CE is and how paradigms in society and industry can be changed to achieve it. The competition also contributed ideas about how society could move toward a more sustainable world and meet the SDGs of the United Nations Agenda 2030. It was held online.

For the implementation of the competition, strategic partnership networks were created for the promotion, dissemination, execution, and awards for the best ideas. Companies from the various sectors of the value chains [5] and potential partner entities outside academia were invited to participate. Some of the entities that institutionally supported the competition are non-profit organisations. All the partners involved (companies, organisations, and institutions) understood the importance of their participation in this project and that their support was an asset to the project, for example, in supporting the dissemina-

tion of a circular economy in academia. In this way, each partner cooperated in the most convenient way for them, such as by helping in the contest promotion and dissemination, supporting the mentoring sessions, and evaluating the project ideas. In addition to the monetary prizes, the partners also offered paid internships for the participating students.

Taking advantage of the established partnership network, the information about the competition was disseminated by email through HEIs on social networks such as Facebook and Instagram, and a website was created (https://gtecrcs.wixsite.com/circularideas (accessed on 31 August 2022)).

Applications for the competition were available via a Google form for 45 days. Participants of the contest and associated teams had to identify the team and the idea (descriptive project) and submit a curriculum vitae. It was recommended that the project content, besides the title and objectives, present: (i) the problem to be solved; (ii) the solution or potential solution, which must include the CE concepts; (iii) the impacts (environmental, social, and economic); (iv) what results are expected from the implementation of the idea; (v) implementation planning; (vi) the business model canvas; and (vii) the framework of the idea for the SDGs. Some critical aspects were considered, such as copyright and publication rights of the awarded works, confidentiality, treatment of personal data, and intellectual property of the ideas submitted.

The assessment of the ideas took place in two phases, the first at the end of April 2021, and the second at the end of June 2021, the final competition, in which three ideas were selected. In the first phase, the five best ideas were selected by a jury of seven members (one from each partner involved and one from the circular economy WG of Sustainable Campus Network). The six partners' evaluators were selected by the organisations and were professionals responsible for implementing a CE in their organisations. The seventh evaluator was from Portuguese academia, had experience in conducting CE research and related expertise, and was a collaborator in the Sustainable Campus Network. All the project ideas that could be implemented on HEI campuses were accepted to the contest. The evaluation of the product or service ideas took into account the criteria contained in the contest regulation, such as creativity and innovation, development, feasibility and operability, and framework in the SDGs of the 2030 Agenda. First, there was an individual evaluation by each evaluator, after which a consensus reunion took place to find the final results.

### 2.1.2. Student Population and Thematic Project Areas in the Competition

The student population was analysed for gender and study area. The study area was based on the submitted students' curriculum vitae.

The thematic areas of the projects submitted to the competition were analysed based on the methodology suggested in Ellen MacArthur's toolkit [22], namely by identifying the most relevant sectors of the economy for the application of the ideas and strategies. The economic sectors were those identified in the final report published by the European community in 2019 [23], such as construction, waste processing, food and feed, manufacturing, mobility, education, energy and heat, agriculture and forestry, chemicals (include plastics), water, processing and management, clothing and textile, electronics, mining, and metals and minerals. Each team framed the project's ideas in the SDGs, which was also evaluated.

### 2.2. Evaluation and Translation to CE Practices (Steps 3 and 4)

The evaluation of the contest projects was based on the British Standards Institution (BSI) [24] and Rossi et al. [25]. The BSI intends to help organisations and individuals consider and implement more circular and sustainable practices within their businesses [24]. Therefore, they developed a framework and guidance for several organisations of differing sizes and levels of knowledge and understanding of the circular economy. This framework intends to be flexible, is based on the CE principles, and provides guidance on mechanisms and business models to support organisations in the transition to a CE [24]. Rossi et al. [25] proposed a set of multidimensional indicators organisations could use to measure CE

performance considering a business model perspective and sustainability based on the three dimensions of sustainable development: environmental, economic, and social. Therefore, in the present work, multidimensional indicators were used to evaluate and determine the knowledge of the contest participants on the CE principles, circular business models, and the pillars of sustainability.

For each project, the contest evaluators verified the circular business models and CE principles used/applied in the accepted projects. Circular business models are presented in Table 1. Each evaluator replied to the question, 'Does the project include one of these models? (Yes/No)'. For the CE principles, in Table 2, the question was, 'Does the project include CE principles? Which ones?'.

**Table 1.** Description of the circular business model used in this work, adapted from [24].

| Business Models | Description |
| --- | --- |
| On demand | Produce on demand (made to order) |
| Dematerialisation | Digitisation |
| Product life cycle extension/reuse | Product life extension, facilitated reuse, product modular design, refurbish, repair, remanufacture, and recondition |
| Recovery of secondary raw materials/by-products | Recovery of secondary materials/by-products (including recycling), incentivised return/extended producer responsibility |
| Products as services/ product–service system | Lease agreement, performance-based (pay for success) |
| Sharing economy and collaborative consumption | Sharing economy, sharing platforms/resources |

**Table 2.** Description of the CE principles used in this work.

| CE Principles | Description | Adapted From: |
| --- | --- | --- |
| Systems thinking | The project integrates parts of a system to produce the behaviour of the whole. | [24,26] |
| Innovation | The project creates value through the design of processes, products/services, and business models for sustainable management of resources. | [24,27] |
| Stewardship | The project manages the direct and indirect impacts of its activities within the wider system that it is part of. | [24,28] |
| Collaboration | The project includes collaboration between stakeholder chains to create mutual value. | [24,29,30] |
| Value optimisation | The project keeps all products, components, and materials at their highest value and utility at all times, optimising every aspect of a product's life cycle. | [24,31] |
| Transparency | The project is willing to communicate these in a clear, accurate, timely, honest, and complete manner. | [24,32] |

The indicators in the three pillars of sustainable development were also used to measure the CE in the tendered projects. The objective was to understand the alignment of the project with the three pillars of sustainable development. The indicators were selected from various authors with work developed in this domain, such as the Ellen Macarthur Foundation [32], Rossi et al. [25], OECD [33], and Padilla-Rivera et al. [34]. The literature showed that most indicators in a CE focus on material flows [28,35]. However, other indicators were included for analysing all sustainability dimensions [36]. Table 3 presents the selected indicators.

**Table 3.** Sustainability indicators selected (from [25,32–34]).

| Sustainable Development Pillar | Indicators |
| --- | --- |
| Environmental | Renewable materials incorporated (%) |
| | Reduction in toxic substances (%) |
| | Material recovered through renewability (%) |
| | Material recovered through recycling (%) |
| | Material recovered through reuse (%) |
| | Material recovered through remanufacturing (%) |
| | Material recovered through refurbishment (%) |
| | Product longevity (years) |
| Social | Training and education activities (capacity building) (yes/no) |
| | Inclusiveness (yes/no) |
| | Social networks involved (yes/no) |
| | Consumer health and safety awareness (yes/no) |
| | No. of actions realised through a platform for the sharing economy |
| | Stakeholder engagement (yes/no) |
| Economic | Waste reduction economic savings (EUR/kg) |
| | Savings as a consequence of recovery and reuse of materials (EUR/kg) |
| | Savings as a consequence of recycling materials (EUR/kg) |
| | Efficiency in resource productivity (EUR/kg) |

For each of the indicators, the contest evaluators assigned a qualitative rating based on the linkage between the expected results from the submitted projects, according to the following scale (adapted from Rossi et al. [25]):

- ● Strong relation: the project can be measured using this indicator and will provide very significant results;
- ◉ Median relation: the project can be measured using this indicator and will provide significant results;
- ○ Weak relation: the project can be measured using this indicator but will provide no significant results;
- ☒ No relation: the project cannot be measured using this indicator.

## 3. Results and Discussion

### 3.1. The Contest Ideas

In the first phase of the competition, 30 ideas were submitted from 144 participants and 20 HEIs located in Portugal and Brazil, and 23 were accepted based on the presented criteria. Of the 23 ideas accepted to the competition, 73.3% were from public institutions, and 70% were from Portuguese HEIs. The study population was mostly from Portugal (64%) and female (54%). A total of 57% students were in science and technology courses, 20% in health sciences courses, 17% in economic and management courses, and 3% in communication and psychology courses (Figure 2a). The gender distribution by study area showed that for science and technology, economics and management, and communication, gender distribution was similar for males and females. However, for health sciences, females accounted for almost 80%, and for psychology, 100% (Figure 2b).

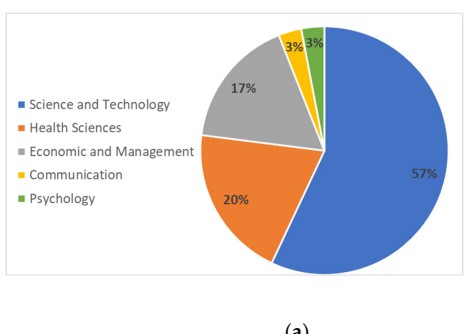

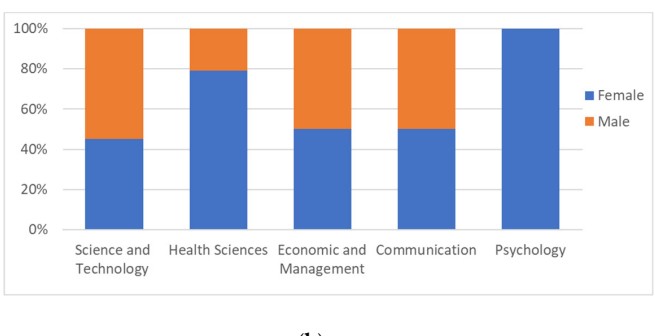

(**a**)                                                 (**b**)

**Figure 2.** (**a**) Distribution of students by areas of study; (**b**) gender of students by area of study.

Figure 3 shows the results for the projects' considered economic sectors. The most common economic sectors used for the development of ideas were related to waste processing (34%), food and feed (13%), education (13%), and agriculture and forestry (10%). None of the projects covered mining, metals and minerals, or mobility sectors.

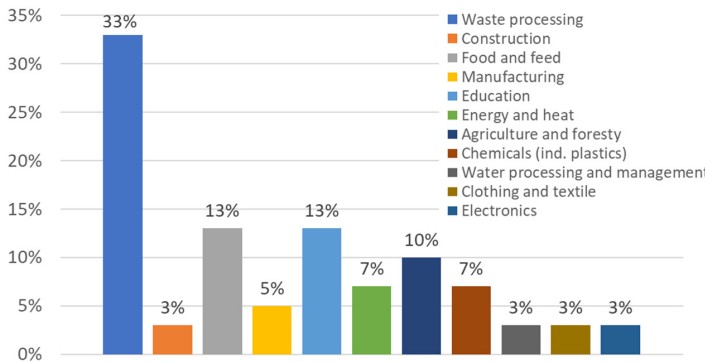

**Figure 3.** Distribution of the project ideas in the contest by economic sector.

The students' study areas and their relation to the projects' economic sectors are presented in Figure 4. The students from the science and technology area presented ideas in all the economic sectors and were very well-represented. Students from the health sciences area presented projects in waste processing, food and feed, manufacturing, education, and chemicals. The least represented student area was psychology, with projects in just one economic sector, manufacturing. There were three economic sectors where only the students from the science and technology area presented projects, namely construction, clothing and textile, and electronics.

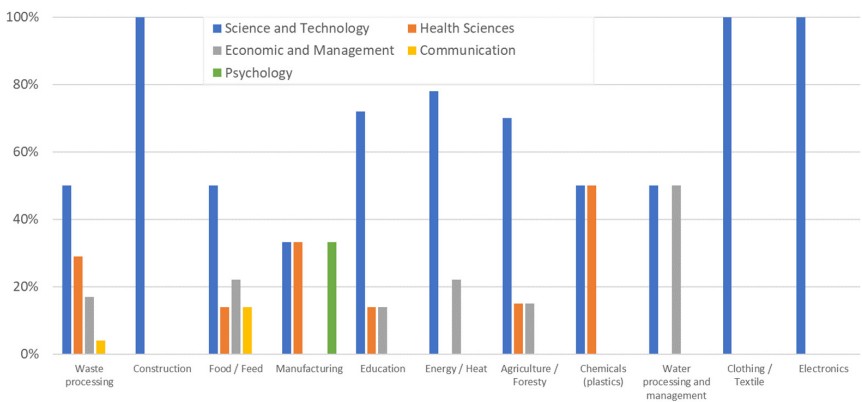

**Figure 4.** Relationship between the study areas of the students and the economic sectors of the projects in the contest.

The analysis of the students' perception regarding the projects' contribution to the SDGs revealed that SDG12 (responsible consumption and production) was the most indicated in 80% of the projects, followed by SDG15 (life on land) with approximately 60%, SDG13 (climate action) and SDG14 (life below water) with ~50%, and SDG17 (partnerships for the goals) and SDG9 (industry, innovation, and infrastructure) with ~40% of the projects. None of the projects covered SDG16 (peace, justice, and strong institutions).

### 3.2. Translation of the Contest Results to CE Concepts and Practices

The results of the contest evaluation are presented in Tables 4 and 5. All the projects accepted were evaluated.

**Table 4.** The number of the contest projects that considered circular business models and CE principles.

| Circular Business Models | No. of Projects | CE Principle | No. of Projects |
|---|---|---|---|
| On demand | 0 | Systems thinking | 4 |
| Dematerialisation | 1 | Innovation | 8 |
| Product life cycle extension/reuse | 4 | Stewardship | 2 |
| Recovery of secondary raw materials/by-products | 11 | Collaboration | 14 |
| Products as services/ product–service system (PSS) | 4 | Value optimisation | 18 |
| Sharing economy and platforms and collaborative consumption | 3 | Transparency | 1 |

**Table 5.** Percentage and type of relationships between the projects submitted and selected indicators.

| | Environmental Indicators (%) | | | | | | | |
|---|---|---|---|---|---|---|---|---|
| | Renewable Materials Incorporated | Reduction in Toxic Substances | Material Recovered Through: | | | | | Product Longevity |
| | | | Renewability | Recycling | Reuse | Remanufacturing | Refurbishment | |
| ● | 22 | 0 | 0 | 48 | 22 | 0 | 4 | 57 |
| ◉ | 9 | 9 | 0 | 9 | 4 | 8 | 0 | 17 |
| ○ | 34 | 70 | 78 | 22 | 52 | 70 | 74 | 13 |
| ⊠ | 35 | 21 | 22 | 21 | 22 | 22 | 22 | 13 |

| | Social indicators (%) | | | | | |
|---|---|---|---|---|---|---|
| | Training/education activities | Inclusiveness | Social networks involved | Consumer health and safety awareness | No. of actions realised through a platform for the sharing economy | Stakeholder engagement |
| ● | 0 | 0 | 30 | 9 | 4 | 43 |
| ◉ | 13 | 0 | 4 | 4 | 13 | 22 |
| ○ | 48 | 0 | 4 | 17 | 0 | 13 |
| ⊠ | 39 | 100 | 62 | 70 | 83 | 22 |

| | Economic indicators (%) | | | |
|---|---|---|---|---|
| | Waste reduction economic savings | Savings due to recovery and reuse of materials | Savings as a consequence of recycling materials | Efficiency in resource productivity |
| ● | 4 | 26 | 39 | 13 |
| ◉ | 35 | 9 | 0 | 13 |
| ○ | 17 | 4 | 9 | 17 |
| ⊠ | 44 | 61 | 52 | 57 |

Results show that most projects' business models were related to the recovery of secondary raw materials/by-products circular business models, with 11 projects in this area (Table 4). There were four projects focusing on product life cycle extension/reuse models,

four on products as services, and three on sharing economy. The dematerialisation circular model included only one project (Table 4). For circular principles, 18 projects included the value optimisation principle, 13 collaboration, 12 systems thinking, and 8 innovation principles (Table 4). Just one project included the transparency principle, and two the stewardship principle (Table 4).

These results are very interesting and are somehow related. Most of the projects' ideas were related to the recovery of secondary raw materials/by-products model, so the value optimisation principle, which assumes that all products, components, and materials at their highest value and utility must be kept, optimising every aspect of a product's life cycle, was the baseline of the projects' ideas when considering a CE. The fact that most of the proposed projects included this circular model and CE principle may represent a misconception of the CE. Webster [37] considered that the first misconception about a CE is 'it is recycling on steroids', which means that a CE is a better way of recycling or waste management. In addition, Kirchherr et al. [38] analysed 114 definitions of circular economies and observed that, after 2014, 72% of definitions included 'recycling', 70% included 'reuse', 50% 'reduce', and 7–9% 'recover'. These authors concluded that scholars must be aware of the differences in the conceptual understanding of CE; otherwise, misleading results may be generated. It is interesting to analyse the number of projects that included transparency, stewardship, and systems thinking principles. Transparency is about the willingness to communicate this in a straightforward, accurate, timely, honest, and complete manner [24], and it was considered that just one project had this principle in mind. For stewardship, this was just two projects. Here, the project should manage the impacts of its activities within the broader system that it is part of [24]. For example, Kalfagianni et al. [39] believed that stewardship is important to highlight deep interconnections between humans and the biosphere. Therefore, this concept is gaining increasing attention from scholars in the field of global environmental change [39]. The system thinking principle, where a project integrates parts of a system to produce the behaviour of the whole system [24], was attributed to four projects. This principle is related to the ability to collectively analyse the complexity of systems across different domains, such as society, the environment, economy, etc., and scales (local to global) [40]. Therefore, it is used to analyse a system and to identify possibilities to change a system to satisfy the needs of a specific group [41].

Table 5 shows that most projects could be evaluated using environmental indicators. The students slightly considered the social and economic indicators of the CE. For environmental indicators, product longevity, which could be applied to measure the results of the projects proposed by the students, was the indicator with the strongest relation (57%) followed by the material recovered through recycling (48%). Material recovered through reuse and incorporation of renewable materials obtained 22%. Very weak relations were observed for projects related to the reduction in toxic substances (70%), material recovered through renewability (78%), remanufacturing (70%), and refurbishment (74%). Looking at social indicators (Table 5), stakeholder engagement and social networks were the indicators with the strongest relations, 43% and 30%, respectively. One of the social indicators, inclusiveness, was not included in any of the proposed projects. Finally, the strongest relations in economic indicators were savings as a consequence of recycling materials (39%) and savings of recovery and reuse of materials (26%), following the obtained environmental indicators.

The prevalence of environmental indicators and the almost non-existence of social indicators corroborate the analysis that students considered the CE to mainly be related to recycling. A CE aims to create economic value through the economic value of materials, social value through the minimisation of social value destruction in the entire system, and environmental value through the resilience of natural resources [42]. This was not the students' thoughts on the projects when they were presented. However, as Kirchherr et al. [38] and Corvellec et al. [43] believed that the CE would become mainstream and move beyond sustainability, a conceptual coherence among definitions, plans, implementations, and modes of evaluation is necessary. These authors concluded that this coherence is funda-

mental for expanding new knowledge on CEs, which may represent a challenge for HEIs. HEIs can identify best practices for implementing CEs. Serrano-Bedia and Perez-Perez [18] found that the initiatives developed by HEIs for CEs were focused on curricula reforms. However, despite these efforts, the results from this study demonstrate that there is still a way to go. HEIs need to change curricula to enrich college courses with transdisciplinary CE competencies. HEIs should include strategic circularity-related competencies to develop the circular business model and product design concurrently and to anticipate how the circular offering will evolve over multiple life cycles [44]. According to Janssens et al. [45], transversal competencies and valorisation competencies are equally important as technical competencies for a CE implementation.

## 4. Conclusions

HEIs have the core responsibility of knowledge dissemination in academia and civil society. The implementation of the first CE competition launched by the CE Working Group of the Portuguese Sustainable Campus Network for Portuguese-speaking HEI students in 2021 is an example of an action to empower students' understanding of CE concepts and sustainability in its various dimensions. An evaluation of students' knowledge of CE concepts and principles and their contribution to sustainability in HEIs was performed in this work.

Of the 23 ideas accepted, mainly from Portugal and public HEIs, ca. 60% of the students came from the science and technology area, and 20% came from both health sciences and economic and management areas. These main students' study areas influenced the economic sectors considered in the projects presented, mostly including waste processing, food and feed, education and agriculture, and forest economic sectors. Nevertheless, the projects based on waste processing represented more than 30% of the rest of the sectors. The relationship between the distribution of the students' fields of study and the economic sectors of the projects showed that science and technology students contributed most of the proposals.

This study showed that the recovery of secondary raw materials/by-products was the CE business model most frequently applied in the projects (11 projects), and value optimisation was the most frequently considered CE principle (in 18 projects). These results may represent a misconception of the CEs because of the students' perception that a CE is only about recycling and waste management. Transparency and stewardship concepts were entirely out of the students' thoughts regarding the projects' CE principles (being considered in one and two projects, respectively). In addition, the results show that the environmental dimension existed in the CE concepts of the projects, but the students only somewhat considered social and economic dimensions. Additionally, the strong relationships between some indicators, such as product longevity, material recovered through recycling, material recovered through reuse, and incorporation of renewable materials, corroborate the students' misconception of the CE concepts and principles and their relation to sustainability implementation. Students considered in their projects that a CE is mainly related to recycling. Therefore, information about CE concepts, principles, and models needs reinforcement and transversal approaches to increase dissemination of knowledge on the subject.

The project's limitations are related to the evaluation and translation of CE practices. The methodology proposed uses a set of evaluators, experts in CE, to evaluate the projects based on objective questions answered in the assessment. The evaluation had two phases: an individual and a consensus reunion with all the evaluators. This methodology was applied to increase the quality of the evaluation process. However, the evaluation could always be improved by clarifying all the steps, for example, in a pre-meeting with the evaluators. This work presents a contest as a methodology that could be used to develop and assess the students' knowledge and understanding of CE concepts and principles and their relation to sustainable development. It is a practical methodology that could help HEIs in the challenge of knowledge dissemination and motivating students to learn,

understand, and implement the transdisciplinary CE competencies. It intends to contribute to extending the value and impact of HEI teaching and research activities and campus initiatives going forward in understanding the CE and its implementation and relation to sustainable development. It also has social implications since HEIs include a wider social context and actors. The understanding of and awakening to the CE in this ambience may ensure, in the future, a positive transition to circularity for work and workers. Additionally, it shows the possibility of increasing cities' liveability, creating employment opportunities, increasing citizens' disposable income, and promoting responsible production and consumption. It will foster the practice of all the knowledge areas of the CE and its relation to sustainable development.

Future research could explore lessons derived from students' participation in CE projects at HEIs, how they understand the concepts and principles, and how they could apply them, as this could be useful in transitioning to circularity. Detailed analysis of the students' perspectives and environmental knowledge could add additional value to the CE transition and should be further investigated in future studies since their contributions to sustainable development will differ based on the education acquired.

**Author Contributions:** Conceptualisation, implementation, and evaluation of the contest, D.D., M.V.B. and C.R.; methodology, C.R. and M.R.T.; writing—original draft preparation, review, and editing, C.R. and M.R.T.; supervision, C.R. and M.R.T. All authors have read and agreed to the published version of the manuscript.

**Funding:** This research received no external funding.

**Institutional Review Board Statement:** Not applicable.

**Informed Consent Statement:** Informed consent was obtained from all subjects involved in the study.

**Acknowledgments:** The partner entities that supported the contest were Delta Cafés, Entogreen, Intraplás, and Soja de Portugal. In addition, other institutions supported the contest as institutional support, such as the World Wildlife Fund—Portugal, Smart Waste Portugal, Circular Economy Portugal, Instituto Politécnico de Castelo Branco, Universidade de Coimbra, and Universidade Tecnológica Federal do Paraná (Brasil). The authors thank the Portuguese Sustainable Campus Network for their technical support of the contest.

**Conflicts of Interest:** The authors declare no conflict of interest.

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
