# Peer review of "From Linear to Circular Ideas: An Educational Contest"

_sustainability, doi:10.3390/su141811207_

Round 1
Reviewer 1 Report
Dear authors,
Thank you for the opportunity to review this work.
Please, see below some comments/suggestions for your evaluation if they can contribute to improve your work.
In the Abstract, it is recommended to provide information on the method. It is possible to understand that the projects submitted to this contest were the source of data, but how this data were extracted from the projects, transformed into “students knowledge” and analyzed?
On page 2, please verify the correct citation of references “…[3], [12]” – There are other in-text citations that need to be corrected, e.g., “Mendoza et al. [3], [15] proposed…”, “[25], [26]”; please verify the manuscript
The Introduction ends abruptly with the article's objective statement. More connection with the literature in the previous paragraph is needed, particularly demonstrating what literature gap exists. The article's contribution to address this gap needs to be made clear.
There are two different objectives stated: “to analyse the Portuguese-speaking HEIs students knowledge about the models, principles and indicators related to the CE concepts and the potential of CE to sustainability, using a contest” and “to evaluate the impact and students knowledge on the CE concepts and its potential to be applied at the HEIs campus of the RCS” – please clarify this to the reader according to what was done. Then, there is a “goal” in Figure 1 also different from the others. Please consider stating the objective of the study clearly. It should be taken into account that the contest is not the research itself, so they can have different objectives, but this is very confuse in the current manuscript.
Please explain what “RSC” means since it does not match “Portuguese Sustainable Campus Network”
It would be recommended to referece the web page of the contest
What is your unit of analysis? Students? Projects? Other?
Please clarify the difference of “university institutions vs. polytechnic institutions”, since there is no standard classification and this is different in countries
Is Figure 2 part of methodology? This question is raised because it does not contribute to understand how the characterization of the students was done, but the outcome of this process.
The authors stated that “BSI [20] framework assists organizations and individuals in more circular and sustainable practices within institutions” – But how it assists organizations and individuals? Based on what it can be state that it assists? Please explain what is BSI and how this contributes to accomplish what is stated.
The images must be presented with better resolution; it is necessary to enable the clear reading of the information
In the section “Results and Discussion”, there is the following sentence: “From the 30 proposals received, only 23 were considered in this study since 7 did not fit into the subject of the contest.” – Why this is not explained in the methodology? The methodology leads to the understanding that the 30 projects were considered. Other questions should naturally be raised: How was the process of selecting the projects for the research conducted? What were the inclusion and exclusion criteria included? How many analysts performed this evaluation? How was the consensus to include/exclude achieved?
Related to that, it is difficult to connect the results from 3.1 and 3.2, since they refer to different data and different samples. Results and conclusions are severely hampered by this disconnect, mainly because the analysis was done considering only percentages.
It is very complicated to accept that "student knowledge" was evaluated based on some project information (theme, SDG, etc.) and percentages calculated from different samples. Simple propositions can overturn this analysis, as students may know a topic, SDG, CE principles or CE as a whole very well, but their project simply did not refer to them. Unfortunately, as a consequence, it cannot be said that the objective of the research was achieved.
Standardization – e.g., Science and Technology área / science and technology área
A review of the English language is highly recommended to improve readability and understanding, as there are many sentences that sound strange, employing unusual expressions and/or verbs. The authors should improve the writing: spelling errors (e.g., “Metodology”, “anlysis”, “pratical”), verbal tense used wrong, use of verbs that may sound strange (e.g., "proclaimed")
No limitations of the study are stated.
Author Response
Comment 1: “In the Abstract, it is recommended to provide information on the method. It is possible to understand that the projects submitted to this contest were the source of data, but how this data were extracted from the projects, transformed into “students knowledge” and analyzed?”
It was included.
Comment 2: “On page 2, please verify the correct citation of references “…[3], [12]” – There are other in-text citations that need to be corrected, e.g., “Mendoza et al. [3], [15] proposed…”, “[25], [26]”; please verify the manuscript.”
We corrected the citation references.
Comment 3: “The Introduction ends abruptly with the article's objective statement. More connection with the literature in the previous paragraph is needed, particularly demonstrating what literature gap exists. The article's contribution to address this gap needs to be made clear.”
Introduction was reformulated to increase the connection between the objective and literature review and demonstrating what literature gap exists. The article contribution was included.
Comment 4: “There are two different objectives stated: “to analyse the Portuguese-speaking HEIs students knowledge about the models, principles and indicators related to the CE concepts and the potential of CE to sustainability, using a contest” and “to evaluate the impact and students knowledge on the CE concepts and its potential to be applied at the HEIs campus of the RCS” – please clarify this to the reader according to what was done. Then, there is a “goal” in Figure 1 also different from the others. Please consider stating the objective of the study clearly. It should be taken into account that the contest is not the research itself, so they can have different objectives, but this is very confuse in the current manuscript.
Yes, in fact the objective is not clear. We agree. Therefore, we reformulated it to clarify. The objective of the work was to evaluate the impact and students’ knowledge on the CE concepts and its potential to be applied at the HEIs campus. Figure 1 was corrected.
Comment 5: “Please explain what “RSC” means since it does not match “Portuguese Sustainable Campus Network”.”
It was corrected.
Comment 6: “It would be recommended to reference the web page of the contest.”
It was made.
Comment 7: “What is your unit of analysis? Students? Projects? Other?”
The unit of analysis is projects. All the information presented is projects except the one presented in Figure 2.
Comment 8: “Please clarify the difference of “university institutions vs. polytechnic institutions”, since there is no standard classification, and this is different in countries”.
We removed this sentence because it is not important for the work.
Comment 9: “Is Figure 2 part of methodology? This question is raised because it does not contribute to understand how the characterization of the students was done, but the outcome of this process.”
Yes, we agree with the reviewer, so we changed to Results and discussion.
Comment 10: “The authors stated that “BSI [20] framework assists organizations and individuals in more circular and sustainable practices within institutions” – But how it assists organizations and individuals? Based on what it can be state that it assists? Please explain what is BSI and how this contributes to accomplish what is stated.”
We reformulated the sentence to clarify.
Comment 11: “The images must be presented with better resolution; it is necessary to enable the clear reading of the information.”
The resolution of the figures was improved.
Comment 12: “In the section “Results and Discussion”, there is the following sentence: “From the 30 proposals received, only 23 were considered in this study since 7 did not fit into the subject of the contest.” – Why this is not explained in the methodology? The methodology leads to the understanding that the 30 projects were considered. Other questions should naturally be raised: How was the process of selecting the projects for the research conducted? What were the inclusion and exclusion criteria included? How many analysts performed this evaluation? How was the consensus to include/exclude achieved?
Related to that, it is difficult to connect the results from 3.1 and 3.2, since they refer to different data and different samples. Results and conclusions are severely hampered by this disconnect, mainly because the analysis was done considering only percentages.”
The sentence is correct, but it is not in the right place. We remove from 3.2 and included in 3.1 reformulating to clarify. All the projects accepted were evaluated. Projects were accepted based on the criterion that the idea presented could be implemented on the HEIs campus. The jury analysed the projects and verified this criterion. This information was included in methodology. The number of the jury members are in the methodology - the jury was composed by 7 members, one each member from the partners that support the contest and one from the CE working group of the Sustainable Campus Network.
We understand the comment about connect the results from 3.1 and 3.2. In 3.1 we wanted to characterise the sample and understand which students are more aware and informed (and show interest) in CE. In 3.2 we analysed the results based on the methodology presented and described. We tried to clarify this.
Comment 13: “It is very complicated to accept that "student knowledge" was evaluated based on some project information (theme, SDG, etc.) and percentages calculated from different samples. Simple propositions can overturn this analysis, as students may know a topic, SDG, CE principles or CE as a whole very well, but their project simply did not refer to them. Unfortunately, as a consequence, it cannot be said that the objective of the research was achieved.”
The sample was always the same, 23 projects. Yes, we understand the remark. However, one of the contest requirements was that projects should include the CE concept as a whole, so we think that the objective of the research can be achieved. We reinforce this in methodology.
Comment 14: “Standardization – e.g., Science and Technology área / science and technology área”
It was made.
Comment 15: “A review of the English language is highly recommended to improve readability and understanding, as there are many sentences that sound strange, employing unusual expressions and/or verbs. The authors should improve the writing: spelling errors (e.g., “Metodology”, “anlysis”, “pratical”), verbal tense used wrong, use of verbs that may sound strange (e.g., "proclaimed").”
We imporved the English.
Comment 16: “No limitations of the study are stated.”
It was included in conclusion section.

Reviewer 2 Report
This paper attempts to discuss an interesting question about the Higher Education Institutions (HEIs) students’ knowledge about the models, principles and indicators related to circular economy (CE) concepts and the potential of CE to sustainability. Overall, this is an interesting research subject. However, I have some concerns that should be further addressed.
1. Some typos need to be fixed. Carefully check the typos in the paper to ensure that they are correct. For example, “students” in the line 1 of the Abstract part might be “student’s”.
2. Introduction section should be strongly strengthened. I suggest that the contributions and research questions of this work should be presented in the introduction section.
3. The authors should update the latest data and literatures in the introduction and literature review sections, such as doi: 10.1016/j.cie.2020.106951. In addition, literature review should not a simple stack of papers, but a comprehensive analysis. I suggest that the author should organize and summarize relevant literatures.
4. Methodology section and Results and Discussion section should be mentioned more briefly and clearly. There are a number of contents and methods in this paper, it is a litter hard to evaluate which were developed by the authors and which parts were taken from published works. Moreover, the application of practical examples should be added to the Results and Discussion section.
5. Conclusion section is weak and it should be should be strongly strengthened. I suggest that the author should provide some managerial insights from the results. Moreover, the practical examples should also be added to the conclusion section.
6. The logic of this paper is not very clear, and the language should be improved.
Author Response
Comment 1: “Some typos need to be fixed. Carefully check the typos in the paper to ensure that they are correct. For example, “students” in the line 1 of the Abstract part might be “student’s”.”
We checked typos and corrected then. Thank you very much.
Comment 2: “Introduction section should be strongly strengthened. I suggest that the contributions and research questions of this work should be presented in the introduction section.”
The contributions and research questions were included in introduction section as suggested by the Reviewer.
Comment 3: “The authors should update the latest data and literatures in the introduction and literature review sections, such as doi: 10.1016/j.cie.2020.106951. In addition, literature review should not a simple stack of papers, but a comprehensive analysis. I suggest that the author should organize and summarize relevant literatures.”
When we wrote the literature review, we tried to do a comprehensive analysis by i) introducing the CE concept and challengers, ii) relating the CE with sustainability, due to the importance of education for sustainable development, iii) explaining the contribution of the Higher Education Institutions (HEI) to promote, teach, disseminate and awareness the circular economy concepts and ideas and sustainability, and iv) presenting actions that HEI have been applied to reach iii). Therefore, we considered that the article proposed by the reviewer is not related with our work since it is focus on a closed-loop supply chain for WEEE of electronic consisting of a manufacturer, a retailer and a third-party recycler.
We included new relevant literature trying to go forward the reviewer suggestion.
Comment 4: “Methodology section and Results and Discussion section should be mentioned more briefly and clearly. There are a number of contents and methods in this paper, it is a litter hard to evaluate which were developed by the authors and which parts were taken from published works. Moreover, the application of practical examples should be added to the Results and Discussion section.”
We improved the methodology section as well as the Results and Discussion section to clarify, and as suggested by the reviewer.
Comment 5: “Conclusion section is weak and it should be should be strongly strengthened. I suggest that the author should provide some managerial insights from the results. Moreover, the practical examples should also be added to the conclusion section.”
We tried to strengthen the conclusion and include some results.
Comment 6: “The logic of this paper is not very clear, and the language should be improved.”
We tried to improve the logic of the article and improved the language.

Reviewer 3 Report
Title From linear to circular ideas: an educational contest Major comments: 1. It is not clear from the abstract how CE is related to recycling according to their explanation.
Minor Comments: 1. Need to update the problem of solving grammatical issues such as articles, Voice, prepositions, and some sentence patterns.
|
Author Response
Comment 1: “It is not clear from the abstract how CE is related to recycling according to their explanation.”
It was corrected.
Comment 2: “It is unclear to me from the introduction why they have chosen their topics and what are the lacking’s exists in the recent cited study. And what kind of scientific significance exists in their articles compared to existing studies?”
The introduction was strengthened. We included the gaps in literature review and the scientific significance exists in their articles compared to existing studies.
Comment 3: “It is unclear to me how they selected participants for their study. Mainly which sampling method they have used, and how do they use it? need to clarify in the methodology section.”
The criterion to select the projects was included in the methodology section.
Comment 4: “It is necessary to rearrange the Result and Discussion part briefly with a more logical sequence according to their objective.”
This suggestion was made.
Comment 5: “It is unclear to me how this research may be helpful for the scientific community for future research, the authors can include a future research section.”
It was included in conclusion section.
Comment 6: “Need to update the problem of solving grammatical issues such as articles, Voice, prepositions, and some sentence patterns.”
This suggestion was made.

Round 2
Reviewer 1 Report
Dear authors,
Thank you for the revised version of your manuscript.
I believe that the changes made substantially improved the paper.
There are still some errors that need to be fixed:
1) Writing
There are errors such as in page 12 of the PDF: "The study presents some limitation regrading the evaluation of the projectos by the jury..."
Please proofread the text to correct them.
2) Clarity of figure
In Figure 1, in the box on the left, does the "definition" refer to the "goal" or the "impact and knowledge on CE concepts"?
Also, the box on the left is missing a part.
3) Contest evaluators
The authors do not provide any information about the contest evaluators. The results are tied directly to their knowledge and subjectivity. As a consequence, it is critical to know who they are, i.e., how can the reader know that the evaluation is based on the opinions of CE experts?
Please provide information necessary to characterize the group of evaluators.
4) Limitations
The limitations of the study are many. The only sentence related to the limitations of the study actually says nothing about. This needs to be greatly improved.
5) Implications
The study's implications are not discussed. This also needs to be greatly improved. As a suggestion, it can be considered: social implications, implications for theory and practice.
Author Response
Thank you very much for all the comments. We think that they improved our work.
Comment 1: “1) Writing
There are errors such as in page 12 of the PDF: "The study presents some limitation regrading the evaluation of the projectos by the jury..."
Please proofread the text to correct them.”
We reformulate limitations.
Comment 2: “2) Clarity of figure
In Figure 1, in the box on the left, does the "definition" refer to the "goal" or the "impact and knowledge on CE concepts"?
Also, the box on the left is missing a part.”
We reformulated Figure 1. First the goal of the action must be defined. In this case the goal was evaluate the impact and knowledge of the CE concepts. We hope that now it is clear. Thank you very much.
Comment 3: “3) Contest evaluators
The authors do not provide any information about the contest evaluators. The results are tied directly to their knowledge and subjectivity. As a consequence, it is critical to know who they are, i.e., how can the reader know that the evaluation is based on the opinions of CE experts?
Please provide information necessary to characterize the group of evaluators.”
Six of the evaluators are professionals from the organizations that supported the contest. They were selected by the organizations, and we required that the evaluators should be responsible for implementing the CE in their own organizations. The seventh evaluator was from Portuguese academia, with research and expertise on CE. First, there was an individual evaluation and after that a consensus reunion was made to find the presented results, supported by the CE working group of Sustainable Network. We included all this information on the article, on section 2.2.1.
Comment 4: “4) Limitations
The limitations of the study are many. The only sentence related to the limitations of the study actually says nothing about. This needs to be greatly improved.”
We reformulated the limitations.
Comment 5: “5) Implications
The study's implications are not discussed. This also needs to be greatly improved. As a suggestion, it can be considered: social implications, implications for theory and practice.”
We reformulated implications.

Reviewer 2 Report
The paper has been revised as required, but there are still some problems to be noted:
1. The authors should update the latest data and literatures in the introduction and literature review sections, such as doi: 10.1016/j.cie.2020.106951.
2. Pay attention to check the grammar errors in the paper.
Author Response
Comment 1: “1. The authors should update the latest data and literatures in the introduction and literature review sections, such as doi: 10.1016/j.cie.2020.106951.”
We do not understand why the Reviewer suggests including this article since it is not related with the subject of our work, as we explained. The article’s suggested is about a closed-loop supply chain for WEEE of electronic consisting of a manufacturer, a retailer and a third-party recycler, and our work is about CE in HEIs.
Comment 2: “2. Pay attention to check the grammar errors in the paper.”
It was checked.

Reviewer 3 Report
Thank you the authors for updating their articles according to the suggestion of the first review. Though I have not got all answers to the first review properly. But, overall I am satisfied with the improvement of the article. The present form is enough for publications.
Author Response
Comment 1: “Thank you the authors for updating their articles according to the suggestion of the first review. Though I have not got all answers to the first review properly. But, overall I am satisfied with the improvement of the article. The present form is enough for publications.”
Thank you very much . We tried to go forward your comments and suggestions.

Round 3
Reviewer 1 Report
Some changes were made to improve the work.
The manuscript still needs Extensive editing of English language and style.
There are basic errors such as:
2.1.1. Constest framework